# Medication Decision-Making and the Medicines’ Pathway in Nursing Homes: Experiences and Expectations of Involvement of Residents and Informal Caregivers

**DOI:** 10.3390/ijerph20115936

**Published:** 2023-05-24

**Authors:** Amber Damiaens, Ann Van Hecke, Veerle Foulon

**Affiliations:** 1Department of Pharmaceutical and Pharmacological Sciences, KU Leuven, 3000 Leuven, Belgium; 2Department of Nursing Director, Ghent University Hospital, 9000 Ghent, Belgium; 3Department of Public Health and Primary Care, Ghent University, 9000 Ghent, Belgium

**Keywords:** long-term care, nursing home, person-centered care, medicines’ pathway, resident involvement, informal caregiver involvement

## Abstract

Background: Information on how residents and their informal caregivers are involved in the medicines’ pathway in nursing homes is scarce. Likewise, it is not known how they would prefer to be involved therein. Methods: A generic qualitative study using semi-structured interviews with 17 residents and 10 informal caregivers from four nursing homes was performed. Interview transcripts were analyzed using an inductive thematic framework. Results: Four themes were derived to describe resident and informal caregiver involvement in the medicines’ pathway. First, residents and informal caregivers show behaviors of involvement across the medicines’ pathway. Second, their attitude towards involvement was mainly one of resignation, but variation was noted in their involvement preferences, ranging from minimal information to active participation needs. Third, institutional and personal factors were found to contribute to the resigned attitude. Last, situations were identified that drive residents and informal caregivers to act, regardless of their resigned attitude. Conclusions: Resident and informal caregiver involvement in the medicines’ pathway is limited. Nevertheless, interviews show that information and participation needs are present and show potential for residents’ and informal caregivers’ contribution to the medicines’ pathway. Future research should explore initiatives to increase the understanding and acknowledgement of opportunities for involvement and to empower residents and informal caregivers to take on their roles.

## 1. Introduction

Person-centered care (PCC) is defined as care that involves individuals and their informal caregivers (e.g., relatives) to the extent they desire and that respects a person’s autonomy. Furthermore, it implies that decision-making is guided by individuals’ preferences and goals with regard to their health and life [1].

The benefits of PCC for older patients have been established in earlier research. Likewise, positive effects of PCC have been demonstrated for nursing home residents (NHRs), including a better perception of quality of life (QoL), fewer feelings of boredom and depression, as well as improved communication between NHRs and staff [2]. Likewise, resident engagement through setting care goals and respecting residents’ autonomy improves their physical and mental well-being [3,4,5,6].

The implementation of PCC, however, has been proven to be complex. On the one hand, barriers include the lack of skills and the attitude of healthcare professionals (HCPs) [7,8]. Residents themselves, on the other hand, may have the idea that their HCPs are not open to their questions and requests [9]. Furthermore, NHRs might experience communication difficulties, complicating their involvement in their own care [10]. Besides staff and resident-related barriers, environmental factors are also known to have an impact on the provision of PCC. For example, high work pressure among staff (e.g., the perception of having too much to do, having to work at a rapid pace, to have to work overtime) has been shown to be negatively associated with the provision of PCC [11].

Research on PCC with regard to treatment decisions and medication-related processes in nursing homes (NHs) is limited. Current medication-related research in NHs mainly focuses on medicines’ optimization interventions for NHRs, but the majority of these interventions are not person-centered in their approach: residents’ and informal caregivers’ involvement in the medication optimization process remains limited or absent, and the impact or effectiveness of the intervention is most often assessed at the medicines’ level (e.g., appropriateness of prescribing) instead of resident level (e.g., QoL) [12,13,14,15]. Little is known about NHRs’ and informal caregivers’ involvement in decision-making about medicines. Perspectives of NHRs on medication deprescribing have been studied to some degree, highlighting a large amount of trust in the GP among residents and, subsequently, a certain passivity towards being involved in decision-making [16,17]. Only one prior study investigated experiences of NHRs with regard to medication decision-making in general, not focusing on deprescribing specifically, and showed that NHRs’ involvement therein was limited [18].

The medicines’ pathway covers but is not limited to medication decision-making. A total of eight processes can be distinguished in this pathway, going from the resident’s (re-)admission to the NH, overprescribing of the medication, to medication administration and monitoring of (side-)effects [19]. Previous research showed that autonomy with regard to the medicines’ pathway in nursing homes is limited among NHRs [20,21]. Although some countries, including Australia and the UK, provide guidelines on medication self-management (i.e., storage and administration) in residential care facilities, attempting to promote resident independence and autonomy [22,23], research on the implementation of such guidelines is lacking. Moreover, it is not known how NHRs and informal caregivers, who often are a source of support for older adults with regard to managing medications at home [24], would prefer to be involved in the different processes of the medicines’ pathway in the NH.

To meet residents’ and informal caregivers’ expectations and preferences regarding their involvement in the medicines’ pathway after moving into a NH, an understanding of their experiences and expectations with regard to their involvement in the medicines’ pathway, including medication decision-making, is needed. Therefore, the following research questions were formulated for the current study:How do NHRs and informal caregivers perceive their current involvement in the medicines’ pathway and in medication decision-making?What opportunities do NHRs and informal caregivers perceive to be (more) involved in the medicines’ pathway and in medication decision-making?What are contributing factors to the involvement of residents and informal caregivers, or their perception of opportunities to be involved, in the medicines’ pathway and in medication decision-making?

## 2. Materials and Methods

This study was part of a project that was set up in Belgium (RESident’s Participation in the Evaluation and Customization of Therapy, RESPECT), which aims to explore opportunities for NHRs and informal caregiver involvement in the medicines’ pathway and medication decision-making in NHs. Interviews were performed in parallel with NHRs and informal caregivers (current study) and with HCPs involved in the medicines’ pathway in NHs. As such, study methods of the current study were largely similar to those of the interview study performed with HCPs, which has been published elsewhere [25]. Reporting of the study was guided by the Consolidated Criteria for Reporting Qualitative Research (COREQ) [26].

### 2.1. Design

A qualitative, explorative study was executed by means of semi-structured interviews across Flanders, the Flemish-speaking part of Belgium, and Brussels.

### 2.2. Participants

Nursing homes were purposively invited to participate based on three criteria: ownership status (i.e., private nonprofit, private for-profit, or public), number of beds (i.e., 35 to 80 beds, 81 to 150 beds, and more than 150 beds) and location (i.e., one in each Flemish province and one in Brussels). In total, four NHs participated in the interview study [25].

Inclusion criteria for NHRs and informal caregivers were Dutch or French speaking and being able to express experiences and views with regard to their involvement in the medicines’ pathway. No (other) inclusion or exclusion criteria were applied. Nursing home residents were approached by a local study liaison (i.e., member of NH staff), who assessed each potential participant for inclusion and subsequently provided the research team with an overview of the availability of residents that agreed to participate. Upon receipt of this overview, the research team and study liaison agreed on one or more dates to perform the interviews in the NH. For informal caregivers who agreed to be contacted, the study liaison provided their contact details to the research team. Subsequently, a member of the research team contacted each of these informal caregivers to agree on an interview date and location.

### 2.3. Data Collection and Analysis

The interview guide was developed according to the framework of the medicines’ pathway, as previously described by Strauven et al. [19,25]. This framework describes the pathway as an entirety of eight processes: (re-)admission to the NH, medication prescribing, purchase and ordering, delivery, storage, preparation, administration, and monitoring of (side-)effects. Open-ended questions were used to encourage residents and informal caregivers to share their experiences and perspectives with regard to their involvement in each of these processes. Special attention was given to medication decision-making, a specific part of the prescribing process. The prescribing process covers but is not limited to initiatives towards appropriate prescribing for NHRs (e.g., use of a therapeutic drug formulary, interdisciplinary medication review). It also includes the organization and execution of general practitioners’ (GP) consultations, as well as the generation and validation of the drug prescription by the GP and pharmacist [19].

Face-to-face interviews were performed with residents and informal caregivers independently and were supported by pictures of recognizable activities of each process of the medicines’ pathway to clarify the content of these processes. Interviews were performed by A.D., L.F., E.J., P.E., a team consisting of both inexperienced and experienced female researchers, each with a background in pharmacy (i.e., pharmacy students and pharmacist researchers). Unexperienced interviewers followed a course on qualitative research methods and were closely coached by the more experienced interviewers. Furthermore, pilot interviews were performed to discuss interview style and quality.

Interview locations included (calm) communal living areas in the NH and residents’ rooms. One interview with an informal caregiver was performed via Zoom.

Interviews were audio-recorded, which subsequently facilitated the ad verbatim transcription. Transcripts were stored on a password-protected computer and were reviewed for accuracy while listening to the recordings.

Data collection and analysis of the interviews were performed in an iterative process. Thus, themes identified in early interviews became probes for later interviews. Data collection was performed until data saturation was reached (i.e., no new themes emerged).

Data analysis was performed by an interprofessional team consisting of female researchers with a background in pharmacy and nursing (A.D., V.F., and A.V.H.) and by applying an inductive thematic framework analysis [25,27].

### 2.4. Rigor

The research team applied different approaches to ensure study rigor, including careful documentation of study procedures, investigator triangulation, data source triangulation, and regular team meetings to discuss study findings [25]. Team meetings also reduced researcher bias by securing researcher’s reflexivity.

### 2.5. Ethical Considerations

The Ethics Committee Research UZ/KU Leuven (S62647) approved the study, which was conducted in accordance with the principles of the Declaration of Helsinki. Before the start of each interview, written, informed consent for participation was collected. One interview was performed through Zoom, for which informed consent was collected through e-mail before the interview took place.

## 3. Results

Seventeen NHRs and ten informal caregivers were interviewed. Interviews with NHRs lasted 23 to 56 min, with an average of 36 min, with the exception of one interview that was concluded after 16 min because the resident became unwell. Interviews with informal caregivers lasted 28 to 96 min, with an average of 55 min. Table 1 summarizes the characteristics of interviewed NHRs and informal caregivers.

Four themes were derived from the interviews: (1) behaviors of involvement by residents and informal caregivers across the medicines’ pathway, (2) residents’ and informal caregivers’ attitudes towards involvement in the medicines’ pathway, (3) factors that contribute to residents’ and informal caregivers’ attitude, and (4) situations that drive residents and informal caregivers to act, independent of their attitude towards involvement. The themes are described hereafter, supported by illustrative quotes.

### 3.1. Behaviors of Involvement: Initiatives by Residents and Informal Caregivers across the Medicines’ Pathway

When asked about their involvement in the medicines’ pathway, residents and informal caregivers mentioned the intake of (oral) medicines as their sole contribution to the medicines’ pathway (i.e., the process of administration).


*“There’s nothing left other than taking in [the medicines] myself. There’s nothing else.”*
—resident 1, 89 years old.

Although residents and informal caregivers did not acknowledge any other involvement in the medicines’ pathway, they provided examples showing behaviors of involvement in different processes of the resident’s medicines’ pathway, with the exception of the (re-)admission process (see Figure 1).

First, examples showed that residents and informal caregivers contribute to the process of prescribing and, more specifically, to medication decision-making. Although both groups believed not to have a say in this, NHRs and informal caregivers indicated sharing medication-related concerns with their HCPs, including the GP, that could potentially impact medication decision-making. Two situations were deduced that concerned residents: high numbers of medicines and the use of specific medication classes (e.g., antibiotics, morphine, and cortisone). Although interviews showed that these medication-related concerns were prevalent among residents and informal caregivers and often shared with HCPs, it seemed as if these were not always acknowledged by the resident’s GP or NH staff.


*“I cannot handle that very well, antibiotics. Then you cannot eat anymore. That is difficult. Antibiotics, I do not like them. He [GP] knows that. I ask him, ‘isn’t there anything else?’ But then he just says, ‘no, you have to take them’.”*
—resident 5, 81 years old.

Other examples of involvement in the prescribing process included residents and informal caregivers asking questions about medication, suggesting medication changes, or sharing information about the resident’s health (i.e., processes of prescribing and monitoring). Although residents nor informal caregivers acknowledged these initiatives as contributions to the medicines’ pathway, interviews showed that they often indirectly (i.e., following the sharing of an observation) or directly (i.e., following a suggestion) influenced the decision to start, change or stop one or more of the resident’s medicines.

Second, interviews showed involvement in the process of purchase and ordering. Some informal caregivers declared to independently purchase medication or supplements for the resident and subsequently hand over these products to their family member or the NH staff, for example, when a product appeared to be unavailable at the NH’s delivering pharmacy.


*“She [ophthalmologist] said that it would be interesting if she [resident] could continue using these food supplements. But these are not available anymore in Belgium. I found them in France. And now I go and get a box [in France] every 3 months.”*
—informal caregiver 8, resident’s nephew, 64 years old.

Third, some residents mentioned they store (part of) their medication in their own room, with paracetamol being the most frequently mentioned example in this regard (i.e., the process of storage).

Finally, residents and informal caregivers indicated performing visual control on the medication they receive during administration rounds by checking the number and visual properties (i.e., shape and color) of tablets (i.e., the process of administration). While interviews showed that this form of involvement could be seen as a systematic and continuous contribution of residents, informal caregivers executed this form of involvement more sporadically (i.e., when they were present during administration rounds).


*“I need to check that [medication during administration rounds] because it sometimes happens that there is one too many, or one too few.”*
—resident 3, 85 years old.

Informal caregivers also indicated sharing information with NH staff in order to facilitate the resident’s medication administration, which seemed to occur more frequently for residents with cognitive impairment or dementia (i.e., the process of administration).

### 3.2. Attitude towards Involvement in the Medicines’ Pathway: Mainly One of Resignation

Although interviews showed different behaviors of involvement across the medicines’ pathway, residents and informal caregivers did not acknowledge these initiatives as being involved. Furthermore, in general, interviews showed that residents and informal caregivers took on a resigned attitude toward their involvement in the medicines’ pathway or opportunities to possess more responsibilities in this regard. Residents and informal caregivers mentioned being ‘okay’ with their current level of involvement, indicating it is easy and they do not have to worry about the management of medicines. Likewise, the majority of residents and informal caregivers showed resignation towards medication-related decisions being made by the resident’s GP. Both groups generally accepted the resident’s medication plan and did not question decisions made by others in this regard. Both parties believed the GP to prescribe and continue a medication only if necessary.


*“What the physician prescribes, I’ll take and that’s that. I do not ask for an explanation. The physician decides. I do not bother at all. I do what the physician and nurse tell me.”*
—resident 9, 91 years old.

Although the overall attitude could be considered as resigned, variation was noted between individual residents and informal caregivers regarding their level of involvement in the medicines’ pathway and satisfaction therewith. In most cases, involvement seemed to be limited to being provided with information on the medicines’ pathway or medication decisions (i.e., medication changes).


*“The head nurse checked with the treating physician and said to give the standard treatment [for this specific problem]. That’s all. They don’t have to ask me what to do. They inform me. I’m always informed, with perfect punctuality. I’m satisfied for sure. I couldn’t be better informed, I think.”*
—informal caregiver 10, resident’s partner, 83 years old.

While some residents and informal caregivers seemed to be satisfied with a minimum of information provided by the GP or NH staff, others were not and expressed a stronger information need (e.g., the price, potential side-effects of the medication, and duration of treatment). Few residents and informal caregivers also expressed a preference to be more actively involved and participate in medication-related activities (e.g., storage) and decision-making.


*“I would like to be more involved, especially at the level of information. Being informed about what [medicines] they are prescribing, what the aim is of using these [medicines], and what the global strategy is at this moment and eventually tomorrow. Because knowing the theoretical progress of the disease [dementia], you are anxious about the actual situation but also about the future. You wonder if there will be a means to relieve, to soften [the disease], etc. Will there be a medicine available at that moment? It’s clear that I would like to be more informed. There is also the question with regard to the administration of medicines: the resident, does he take his medicines? And at what frequency?”*
—informal caregiver 9, resident’s partner, 68 years old.

### 3.3. Attitude Contributors: Organizational and Personal Factors

Multiple organizational and personal factors were found to contribute to the overall resigned attitude of residents and informal caregivers toward being involved in the medicines’ pathway. The first factor that seemed to contribute to the resigned attitude of residents was the fact that NHs were perceived more as healthcare institutions than as home replacement settings. In this regard, residents seemed to think that an expansion of their involvement across the medicines’ pathway would disrupt the existing work practices of NH staff and therefore considered it to be practically impossible.

Likewise, the institutional perception fostered concerns about patient safety. Although some residents believed they would be capable of taking up more medication-related responsibilities across the medicines’ pathway themselves, they did not seem to expect this opportunity to be provided to them because the majority of NHRs are cognitively impaired and thus not capable of doing so. Informal caregivers did not see opportunities to become more involved in the resident’s medicines’ pathway since they were not present at all times.


*“Too many mistakes would happen. If everyone has their medication in their cupboard [in their room], and someone else enters and takes it… That would lead to serious incidents.”*
—resident 8, 73 years old.

Residents and informal caregivers had no idea about what activities the medicines’ pathway entails, which further limited their perception of (potential) contributions and opportunities to be involved therein. Hence, this lack of knowledge was identified as a second factor contributing to the resigned attitude of residents and informal caregivers.

Another factor that was seen to foster the resigned attitude toward involvement in the medicines’ pathway was residents’ and informal caregivers’ perception of their own cognitive capabilities, or lack thereof. This perception was mostly driven by the high numbers of medicines used, their perceived medication-related knowledge, and the use of professional language during decision-making conversations among HCPs.

Last, a high degree of trust of residents and informal caregivers in the resident’s GP and NH staff was noted, which was identified as an important contributor to their attitude of resignation towards involvement in the medicines’ pathway and, more specifically, in medication decision-making.


*“I trust my GP to do her work well and that she takes responsible decisions. I cannot control these decisions, nor is that up to me. I don’t have that knowledge, so I have to rely on the physician. And I do that.”*
—informal caregiver 5, resident’s daughter, 64 years old.

### 3.4. Drivers for Action: Time to Speak Up

Despite the resigned attitude of residents and informal caregivers towards (more) involvement in the resident’s medicines’ pathway, interviews showed that both parties take action in certain situations, regardless of their resigned attitude. An example of such a situation is when the continuity of the resident’s medicines’ pathway seems to be threatened. Residents and informal caregivers indicated that receiving tablets or capsules in shapes or colors they do not recognize, as well as discrepancies in the numbers of medicines they receive (i.e., one too many, one too few) during administration rounds, encourages them to consult a member of the NH staff. Moreover, residents and informal caregivers considered themselves responsible for being attentive during administration rounds and checking for potential errors.


*“It happened once, many months ago, that I received the medicines for my neighbor whom I know well. I simply rang the bell and told that these were not my medicines but those of the woman across [the hall].”*
—resident 13, 91 years old.

Additionally, with regard to medication decision-making, certain situations were derived from the interviews that made residents and informal caregivers take the initiative and voice their thoughts. The first driver for residents and informal caregivers to speak up was the resident experiencing discomfort (e.g., pain). Residents and informal caregivers indicated to share such observations about the resident’s health and health-related complaints to the resident’s GP or the NH staff, which would then potentially result in a change of the resident’s medication plan.


*“There was a period she complained a lot about [pain in] her back and knee. A few days passed. I think the nurses waited some time to see how bad the pain was. At a certain moment, I came in and said, ‘now we have to do something’. I then texted the GP to tell what was going on, that my mom was in a lot of pain and was only receiving half a morphine patch and paracetamol at that time. [I asked] if we could add something like an anti-inflammatory or Brufen [ibuprofen] so she would get some additional pain relief because at that time, it was not enough and she was not comfortable.”*
—informal caregiver 6, resident’s daughter, 55 years old.

Besides this, some residents and informal caregivers described examples of concrete suggestions regarding the resident’s medication, which were driven by the perceived effectiveness or necessity of the medicine in question (e.g., a resident suggesting stopping the prescription of pantoprazole when she no longer experienced gastric discomfort).

## 4. Discussion

While limited research has been carried out on the involvement of NHRs in medication decision-making [16,17,18], our study shows that the involvement of NHRs, but also informal caregivers, is not limited to this single process of the medicines’ pathway. Behaviors of involvement of NHRs and informal caregivers also occur in other processes of the medicines’ pathway, such as the storage of medication, purchase and ordering of medication, and monitoring of medication (side-)effects. Remarkable is the fact that NHRs and informal caregivers do not seem to be involved in the process of (re-)admission to the NH, a crucial timepoint to collect information on the residents and their medicines’ use. Personal preferences of NHRs and informal caregivers regarding medication (de-)prescribing, medication storage, medication administration, etc., may differ. Thus, it is important to assess these preferences and concerns if they are to be implemented and taken into account once admitted to the NH. When compared to the number and extent of activities that the medicines’ pathway contains, study findings show that the involvement of NHRs and informal caregivers in this pathway is limited [19].

Moreover, examples of involvement provided by NHRs and informal caregivers show that their involvement in the medicines’ pathway is often self-initiated. These initiatives, including residents and informal caregivers asking questions about their medicines, suggesting medication changes, and sharing medication-related experiences and concerns, are potential cues for their interest to be involved and implies that, in general, information, as well as participation needs, are present among both groups. Important to note is that the extent to which NHRs or informal caregivers wish to be involved in the medicines’ pathway varies from minimal information needs (e.g., to be informed about medication changes) to more extensive information and participation needs (e.g., to possess medication-related responsibilities and participate in medication decision-making). This variation in involvement preferences among individuals with regard to medical decision-making is confirmed by earlier research [28,29,30]. Nevertheless, study findings show that this is also the case for processes of the medicines’ pathway, other than medication prescribing.

Findings also show that NHRs and informal caregivers, regardless of their individual information and participation needs, do not necessarily expect to be provided with (more) medication-related responsibilities or the opportunity to contribute (more) to medication decision-making. This lack of expectations seemed to hinder certain NHRs and informal caregivers from speaking up and expressing their information and participation needs regarding involvement in the medicines’ pathway. Knowing that the actions of HCPs are reactive and unstructured may cause information and participation needs to remain under the radar [25]. Earlier research already highlighted this attitude of resignation and reconciliation of NHRs towards involvement in the medicines’ pathway [18], but our findings show that this attitude also, in general, characterizes informal caregivers. Interviews highlight several factors that influence this general attitude of resignation and reconciliation, such as the lack of knowledge regarding the processes of the medicines’ pathway. As stated by the Ladder of Patient and Family Engagement by Kim et al., the first step towards involving NHRs and informal caregivers is to inform them about the medicines’ pathway in the NH [31]. Not knowing what activities the medicines’ pathway entails naturally hinders NHRs and informal caregivers in their perception of opportunities to be involved therein. Besides this, the trust in the resident’s GP and NH staff and also the perception of their own (limited) capabilities are factors clearly tempering the expectations of NHRs and informal caregivers with regard to their involvement in the medicines’ pathway. Similar barriers have been described in older adults residing at home, but only recently with regard to medication-related processes in NHs [16,17,30,32,33]. Adding to this, our findings highlight the impact of the perception of an NH as an institution that exists among NHRs and informal caregivers. This perception relates to the medical paradigm that persists in NHs, having medical standards, resident safety, and working routines dominate over resident choices [34]. Of course, routine is needed to organize and facilitate daily work in NHs, but the challenge lies in finding the ideal balance between applying routines that facilitate efficient work practices and patient safety and the provision of holistic, person-centered care, which includes involving NHRs and informal caregivers to the extent each individual desires. Healthcare professionals, too, struggle with balancing such organizational concerns against involving NHRs and informal caregivers more extensively in the medicines’ pathway and medication decision-making [25].

Lastly, our findings highlight the existence of several drivers for action, reflecting certain situations in which NHRs and informal caregivers take action and speak up, regardless of their ‘general’ attitude towards being involved. Examples of drivers are the perceived (non)necessity or (in)effectiveness of medications. This implies that NHRs and informal caregivers do sometimes question the resident’s medication plan, which is not always unjustified, seeing the high prevalence of potentially inappropriate prescriptions in NHRs [35]. Other drivers for action relate to the continuity of the resident’s medicines’ pathway, or potential lack thereof, and show that NHRs and informal caregivers contribute to the interception of medication errors, which have been shown to be highly prevalent in NHs as well [36]. These findings imply that resident and informal caregiver involvement in the medicines’ pathway is a potential strategy to support patient safety in NHs, and that involvement in medication decision-making might be a new approach to optimize medication plans.

Further research should explore initiatives to raise awareness of involvement opportunities for NHRs and informal caregivers. The first step, according to the Ladder of Patient and Family Engagement by Kim et al., is to inform both groups on the processes and activities of the medicines’ pathway, including the prescribing of medication (i.e., medication decision-making). Following this, NHRs and informal caregivers should be empowered to take up their role in the medicines’ pathway to the extent they desire [31]. Patient empowerment leads to a greater sense of self-efficacy and self-management skills among individuals with chronic conditions [37] and thus might be a means to respond to NHR and informal caregiver-related barriers to their involvement in the medicines’ pathway (i.e., perceived lack of capabilities). An example of such an informative and empowering intervention to improve awareness among NHRs and informal caregivers on an individual level is an informed and empowering conversation upon admission. On an organizational level, educational sessions for staff, but also for NHRs and informal caregivers on the processes of the medicines’ pathway and involvement opportunities therein may be helpful.

Indeed, initiatives should also target HCPs involved in the medicines’ pathway of the NH. As interviews with NHRs and informal caregivers show, the information and participation needs of NHRs and informal caregivers are not always acknowledged by their HCP. Interviews with HCPs, investigating their experiences and perspectives on resident and informal caregiver involvement highlighted a similar finding and indicated that the majority of actions towards the involvement of NHRs and informal caregivers by HCPs have a reactive and unstructured nature [25]. Likewise, the current interviews with NHRs and informal caregivers indicate that their involvement in the medicines’ pathway is often self-initiated and that their medication-related concerns are not always considered by HCPs during medication decision-making, even when expressed in a direct manner. Considering NHRs’ and informal caregivers’ experiences and concerns, and involving them in the resident’s care to the extent that they desire, however, are key to PCC [1]. Therefore, efforts should be made to support HCPs in taking up their responsibilities in the provision of PCC and the involvement of NHRs and informal caregivers in the medicines’ pathway and medication decision-making and encourage a more proactive approach from their end to empower NHRs and informal caregivers.

### Strengths and Limitations

Some strengths and limitations need to be addressed. Although not all interviewers were equally experienced, steps were taken to reduce the impact thereof on the quality of the interviews (see Section 2.3). Analysis of the interview data was performed by an interprofessional team, ensuring a holistic approach to the data as well as investigator triangulation. Moreover, this enhanced the rigor and reliability of the study findings and improved researchers’ reflexivity. Furthermore, interviews were performed until data saturation was reached (i.e., until no new themes emerged from the interviews).

The main limitation of the study is that the recruitment of participants was performed by NH staff, which might have resulted in a selection bias. Nevertheless, it became clear during the interviews that a variation of profiles was included with regard to the resident’s or informal caregiver’s current level of involvement in the medicines’ pathway, as well as their preferred level of involvement.

Furthermore, a significant portion of NHRs is diagnosed with dementia (i.e., more than one-third in Belgium), and the number is only increasing. As this group was not included in this interview study, further research should also target these NHRs [38]. Likewise, only a small number of informal caregivers related to residents diagnosed with dementia were questioned. As such, study findings might be different when questioning this population with regard to their wish to be involved in the medicines’ pathway in NHs.

## 5. Conclusions

Involvement of NHRs and informal caregivers in the medicines’ pathway and medication decision-making in NHs is limited and often restricted to the actions they self-initiate across the pathway. In general, NHRs nor informal caregivers seem to expect the opportunity to possess more medication-related responsibilities or contribute more extensively to medication decision-making. This absence of expectations is fostered by NHRs’ and informal caregivers’ perception of an NH as an institute, a lack of knowledge regarding the medicines’ pathway, the perception of one’s own capabilities to be involved therein, and the resident’s and informal caregiver’s trust in the GP and NH staff. Despite this general attitude of resignation, NHRs and informal caregivers expressed information and participation needs to varying degrees. Moreover, certain situations exist in which NHRs and informal caregivers take action, regardless of their resigned attitude towards involvement, showing potential for their contribution to the medicines’ pathway and medication decision-making.

Further research should explore initiatives to improve acknowledgement among NHRs and informal caregivers of involvement opportunities across the medicines’ pathway. Furthermore, empowering initiatives should be implemented to encourage NHRs and informal caregivers to take on their role in the resident’s medicines’ pathway to the extent they individually desire. Lastly, as a significant portion of NHRs is diagnosed with dementia, and these were not the focus of this interview study, future research should also target these NHRs and their informal caregivers.

## Figures and Tables

**Figure 1 ijerph-20-05936-f001:**
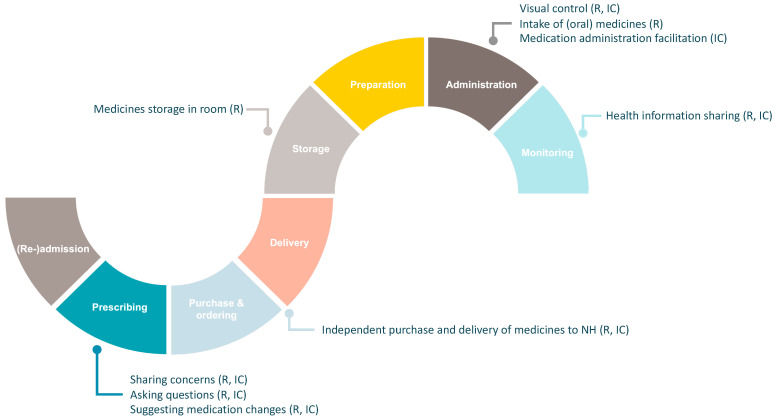
An overview of behaviors of involvement by residents and informal caregivers across the medicines’ pathway (IC: informal caregivers, NH: nursing home, R: residents).

**Table 1 ijerph-20-05936-t001:** Description of nursing home residents and informal caregivers.

	Nursing Home Residents, *N*	17	Informal Caregivers, *N*	10
Age, *n* ^a^				
<60		0		2
61–65		0		4
66–70		2		1
71–75		3		0
76–80		0		0
81–85		7		1
>85		5		0
Gender, *n*				
Female		14		7
Male		3		3
Resident’s number of years since nursing home admission, *n*				
≤1		4		3
2–3		9		2
4–5		2		1
>5		2		4
Resident’s number of chronic medicines used, *n*				
≤5		2		2
6–10		8		5
11–15		4		3
>15		3		0
Relation to resident, *n*				
Partner				2
Child				7
Cousin				1
Resident with diagnosis of dementia, *n*				
Yes				5
No				5

^a^ two missing values for informal caregivers.

## Data Availability

Transcripts analyzed during the current study are available from the corresponding author upon reasonable request. Transcripts are written in the language in which the interview was performed (i.e., Dutch or French).

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
