# Peer review of "Medication Decision-Making and the Medicines’ Pathway in Nursing Homes: Experiences and Expectations of Involvement of Residents and Informal Caregivers"

_ijerph, 2023, doi:10.3390/ijerph20115936_

Round 1

Reviewer 1 Report

Although it is true that the subject is something that has been dealt with many times, the way in which it is treated brings freshness and interesting data, I recommend not using articles in the bibliography in which the authors themselves appear, as this is considered as self-citation.

As he comments in his conclusions, it will be important to create a health education plan for informal caregivers to gain greater autonomy in working with patients' medication.

Author Response

Dear reviewer,

Thank you for your comments.

One paper has indeed been self-cited. However, we chose to keep this in our bibliography as we mainly refer to it in our methods section. These two studies (the one submitted here and the one cited) were performed in parallel and analogously. Hence, to minimize the amount of repetition in this paper (and thus a potential accusation of plagiarism), we refer to the other paper for more details on our data collection and analysis methods. 

Reviewer 2 Report

Title: Medication decision-making and the medicines’ pathway in 2 nursing homes: experiences and expectations of involvement of 3 residents and informal caregivers

Date: 07 March 2023

Reviewer's report

Thank you for the opportunity to review this manuscript. This qualitative study focuses on describing the experiences and expectations of nursing home patients and caregivers regarding their involvement in medication, including the decision-making process.

Overall, I found this paper to be well-written and quite interesting. In my opinion, this manuscript would be of interest to the readership of this journal.

However, I have some doubts and comments that I would like to share with the authors in case they can help improve the quality of the manuscript.

Introduction:

Could the authors address the specific barriers of the nursing home environment to person-centered care and the decision making involved?

Methods:

This study is the second part of a project that was launched in Belgium (RESPECT project). The first part, focused on assessing the attitudes of healthcare professionals on the involvement of residents and informal caregivers in the medication decision-making process in nursing homes, is already published, and the study methods were similar to those of this work. Therefore, they are considered valid and adequate to meet the objective of the study. However, I have some questions and doubts that I would like to raise with the authors:What were the exclusion criteria established?

-          In line 132-133, the authors say: "Interviews were performed by (…)a team consisting of both inexperienced and experienced, female researchers, each with a background in pharmacy". Can you clarify what you mean by " background in pharmacy ", were the interviewers pharmacists?

-          Interview scripts: how was the script developed (i.e, did the authors conduct a literature review to design it? was it based on the experience of the researchers?

-          It also would be good to add the script as an appendix to the article

Results

The authors classified the themes derived from the interviews into four categories (page 5). On what basis were these categories selected?

Discussion

The discussion is well organized and focused. However, the authors could mention some initiatives, both at the individual and organizational level, to raise awareness of this issue. In particular, in view of the results, the help of pharmacists seems relevant to improve the involvement of residents and informal caregivers in medication decision making and in the medication pathway. Certain professional pharmaceutical services should be enhanced: such as medication review, pharmacotherapeutic monitoring, medication reconciliation or adherence : all of them put the patient and his quality of life at the center and focus on the patient and not on the medication. Could the authors discuss about this?

On the other hand, since the authors have conducted a previous study on the topic, could they compare the perceptions of patients and caregivers with those of healthcare professionals to see possible discrepancies and opportunities for improvement? Understanding and taking into account patients' experiences with their medication, as well as the opinions of professionals, is critical to optimize medication decision making.

Bibliography

Last reference: PROBIS. Woonzorgcentra: Trends en indicatoren [Internet]. 2018. Please, add the URL of the reference, and the last time accessed.

Reviewer 3 Report

A very interesting topic, but more data should be added regarding adverse reactions, polypharmacy and types of drugs depending on the pathology.

Author Response

Dear reviewer, 

Thank you for this comment. However, as the focus of the manuscript is on resident/informal caregiver involvement and not on which medicines they take/use, nor on which adverse reactions they experience, we chose to add no additional details on this. We believe that, independent from which medicines a resident is using, he or she should get the opportunity to be involved to the extent he or she desires.  

Reviewer 4 Report

The researchers undertook a qualitative study to examine nursing home resident/caregiver involvement in the medication pathway. Overall, the paper was well written and covered an important topic. The researchers found an overall lack of empowerment among residents and caregivers in the medication pathway despite a desire by many residents/caregivers to be more involved. There are a few areas that could use some additional, minor clarification.

1. The introduction needs a description of what the 8 parts of the medication pathway are. Otherwise, the methods and results are hard to put into context.

2. The researchers claim that triangulation was used to ensure rigor. However, more details are needed as to what is meant. What other data was used to triangulate? 

3. The researchers also held team meetings to ensure rigor. Were these meetings to go over discrepancies in coding among researchers? 

4.  The researchers mention that individuals with cognitive impairment did not participate in the study. Was this an exclusion criteria? Did nursing home staff simply not refer residents with dementia to participate in the study? Either way, I think this should be mentioned in the methods.

 5. I think the researchers can provide additional discussion about the interesting finding that caregivers/residents did not have any role during the admission process. What is the reason for this lack of involvement? What would the role of the resident/caregiver during the admission stage be if the resident/caregiver were empowered?
